

# Optimizing type 2 diabetes management: AI-enhanced time series analysis of continuous glucose monitoring data for personalized dietary intervention

Madiha Anjum[1], Raazia Saher[1] and Muhammad Noman Saeed[2]

[1] Department of Computer Engineering College of Computer Science and IT, King Faisal University, Alahsa, Saudi Arabia

[2] E-Learning Center, Jazan University, Jazan, Saudi Arabia

Corresponding author
Madiha Anjum,
mshahzad@kfu.edu.sa

## ABSTRACT

Despite advanced health facilities in many developed countries, diabetic patients face multifold health challenges. Type 2 diabetes mellitus (T2DM) go along with conspicuous symptoms due to frequent peaks, hypoglycemia <=70 mg/dL (while fasting), or hyperglycemia >=180 mg/dL two hours postprandial, according to the American Diabetes Association (ADA)). The worse effects of Type 2 diabetes mellitus are precisely associated with the poor lifestyle adopted by patients. In particular, a healthy diet and nutritious food are the key to success for such patients. This study was done to help T2DM patients improve their health by developing a favorable lifestyle under an AI-assisted Continuous glucose monitoring (CGM) digital system. This study aims to reduce the blood glucose level fluctuations of such patients by rectifying their daily diet and maintaining their exertion vs. food consumption records. In this study, a well-precise prediction is obtained by training the ML model on a dataset recorded from CGM sensor devices attached to T2DM patients under observation. As the data obtained from the CGM sensor is time series, to predict blood glucose levels, the time series analysis and forecasting are done with XGBoost, SARIMA, and Prophet. The results of different Models are then compared based on performance metrics. This helped in monitoring various trends, specifically irregular patterns of the patient's glucose data, collected by the CGM sensor. Later, keeping track of these trends and seasonality, the diet is adjusted accordingly by adding or removing particular food and keeping track of its nutrients with the intervention of a commercially available all-in-one AI solution for food recognition. This created an interactive assistive system, where the predicted results are compared to food contents to bring the blood glucose levels within the normal range for maintaining a healthy lifestyle and to alert about blood glucose fluctuations before the time that are going to occur sooner. This study will help T2DM patients get in managing diabetes and ultimately bring HbA1c within the normal range (<= 5.7%) for diabetic and pre-diabetic patients, three months after the intervention.

# INTRODUCTION

Diabetes is a chronic medical illness that impairs our body's capacity to convert food into energy. When we eat, most of the food is converted to glucose (also known as sugar) and then released into the bloodstream. Hence, the amount of glucose (sugar) released into our bloodstream rises. This instructs the pancreas to release insulin, which functions as a key to allow blood glucose to enter our body's cells for usage as fuel. Diabetes mellitus consists of several metabolic disorders that affect insulin production, the action of insulin, or both. In patients with diabetes mellitus, insulin is either not produced or unable to attach to the cell to allow glucose to enter. In this case, glucose will build up in the bloodstream. High blood glucose levels can cause significant health issues, such as damage to the heart, blood vessels, eyes, kidneys, and nerves, if left untreated. When we have diabetes, our bodies either do not produce enough insulin or cannot utilize it properly.

Too much blood sugar remains in your bloodstream when insufficient insulin or cells cease reacting to insulin. The International Diabetes Federation (IDF) researched to found that there were 463 million diabetics worldwide in 2019, aged 20 to 79, which is predicted to rise to 700 million by 2045 (*International Diabetes Foundation, 2023*). Since the last 20 years, the number of diabetes diagnoses has doubled, making diabetes the seventh-biggest cause of mortality globally.

According to the American Association of Clinical Endocrinologists (AACE), a dysglycemia-based chronic disease (DBCD) multimorbidity care model consists of four different stages that involve stage 1: insulin resistance, stage 2: pre-diabetes, stage 3: type 2 diabetes mellitus (T2DM) spectrum and stage 4: vascular complications (*Mechanick et al., 2018*). This model encourages initial intervention directed at well-thought-out lifestyle change. Different scientific research may reclassify stage 2 DBCD pre-diabetes into an actual disease state.

While there's no cure for diabetes, managing it through appropriate treatment can help reduce its effect on our bodies and daily routines. In a study by *Norris et al. (2002)*, self-management helps to show immediate better results of glycohemoglobin (Ghb) levels during follow-ups. Once intervention for diabetic management was implemented, the decline was observed within three months after the intervention in diabetic self-management stopped. This shows the need to develop interventions effective in maintaining long-term glycemic control.

The various ways that help in treating may include medications, dietary changes, and lifestyle interventions. Medicines for diabetes can include insulin injections, amylin mimetic drugs, alpha-glycosidase inhibitors, and other drugs to keep blood sugar levels in the normal range. In addition, to help reduce the risk of complications associated with diabetes, many medicines are taken for high cholesterol and high blood pressure and to maintain heart health. There are different tests to screen whether a person is diabetic or pre-diabetic in Table 1, and the criteria (*American Diabetes Association, 2019*) for the screening and diagnosis of pre-diabetes and diabetes are mentioned.

AI has been introduced as an effective and successful technological solution for humanity among the currently available tools (*Cui et al., 2016*). Two main streams of technologies,

**Table 1** **Ranges to diagnose pre-diabetic and diabetic.**

| Tests | Pre-diabetic | Diabetic |
|---|---|---|
| HbA1C | 5.7–6.4 | ≥6.5% |
| FPG (Fasting Plasma Glucose) | 100–125 mg/dL (5.6–6.9 mmol/L[*] | ≥126 mg/dL (7.0 mmol/L |
| OGTT (Oral Glucose Tolerance Test) | 140–199 mg/dL (7.8–11.0 mmol/L)[*] | ≥200 mg/dL (11.1 mmol/L) |
| RPG (Random Plasma Glucose) | – | ≥200 mg/dL (11.1 mmol/L) |

Notes.
[*]In all these tests, the risk is continuous, extending below the lower limit of the range and becoming disproportionately greater at the higher end of the spectrum.

machine learning (ML) and deep learning, emerged from the AI boom in 2021. These have contributed to significant progress due to increased computational resources and considerable improvement in computer performance. Among multiple definitions of AI by the Japanese Society, one of the finest describes that "artificial intelligence aims to accurately make advanced inferences on a large amount of data" (*Japanese Society for Artificial Intelligence, 2023*).

## LITERATURE REVIEW

### AI role in diabetes management

Many studies have been done and are still ongoing to explore ways of managing various diseases in general and diabetes in particular. In diabetes, several machine learning-based medical devices have been introduced, specifically regarding automatic retinal screening, clinical diagnosis support, and patient self-management tools that have already been validated and approved by the US Food and Drug Administration. In the past few years, multiple healthcare apps have been developed for patients to diagnose and manage diseases (*Doupis et al., 2020*; *Martínez-Pérez et al., 2013*), particularly diabetes. However many of these apps have some limitations. First, patients must enter glucose readings in the app every time after checking blood glucose levels measured by a glucometer at home. Second, as dietary control is essential for effective diabetes care, it might occasionally be inconvenient to enter nutritional data repeatedly. However, because it is challenging to look for and enter every detail of food items ingested, at smartphone healthcare apps, patients frequently require assistance in it.

Recently, Bluetooth technology embedded in glucometer has allowed recording blood glucose levels to automatically record readings of patients in any linked smartphone apps (*Saltzstein, 2020*). However, people only occasionally check their blood glucose levels due to the inconvenience of finger-prick testing. In various glucose management applications, the data received through Bluetooth is not integrated with the other required constraints; this may include biometric data such as body weight, blood pressure, water intake, and daily exercise routine. As a result, it is observed that most patients are not in keeping with diabetic management apps on their own, and intervention of medical personnel is required. The key interest of AI in medicine is the construction of such programs that not only assist a medical doctor in performing expert diagnosis and procedures but also improve the quality of life for patients suffering from chronic diseases. By using various computational sciences

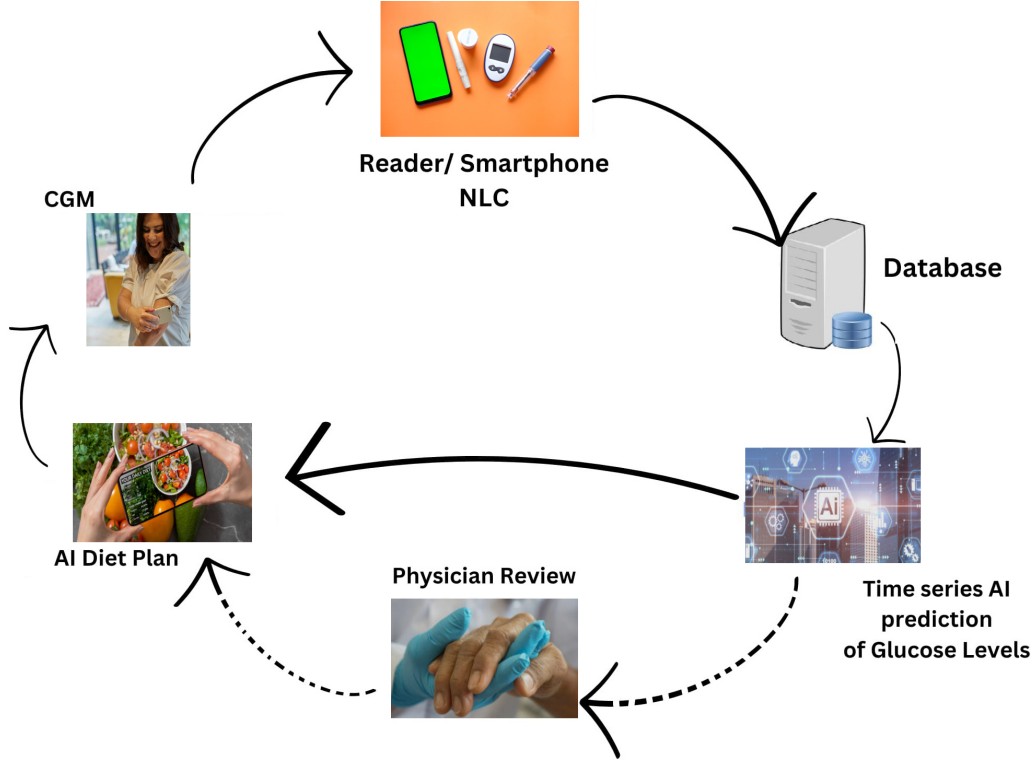

**Figure 1** Revolutionizing type 2 diabetes management: CGM and machine learning algorithm in digital health.

such as statistics and probability, these programs find hidden patterns in the training data, and using these patterns, they classify the test data into one possible category. The support of these programs extends to various data sets, either accumulated from clinical cases recorded in clinics or exported through different healthcare devices in training the system. The decisions and recommendations prepared from these systems can be illustrated to the subjects after combining them with the intelligence of a human expert.

Figure 1, illustrates the comprehensive design of our proposed system, employing an artificial intelligence/machine learning (AI/ML)-driven approach. This system aims to assist individuals with T2DM by facilitating the monitoring and management of their daily dietary habits. The key element of observation revolves around the analysis of their blood glucose (BG) data. In essence, the system leverages advanced AI and ML technologies to comprehensively understand and respond to the intricate relationship between dietary patterns and BG levels in T2DM patients. This approach signifies a proactive and data-driven method for individuals to gain insights into their BG fluctuations, enabling them to make informed decisions about their dietary choices. The emphasis on monitoring and managing daily diet aligns with the broader goal of improving the overall well-being and health outcomes of T2DM patients through the integration of cutting-edge technologies.

The proposed solution aims to address crucial abnormal values in T2DM patients, particularly related to blood glucose (BG) levels. To achieve this, the approach involved

predictive analytics through a combination of time-series analysis with regression analysis on time-series data. The model is built with XGBoost, SARIMA and FBProphet algorithms to forecast future BG values. Considering repeated abrupt fluctuations in our data, XGBoost Regression and seasonal ARIMA performance is compared with the Facebook Prophet. The three models came up with unique outcomes that helped to forecast and estimate factors influencing BG values. The model results are then leveraged to adjust patient diets, with the assistance of a proposed AI-based dietary solution. This comprehensive strategy integrates advanced analytics and artificial intelligence to enable personalized timely adjustments in diet based on predicted BG values. Therefore, potentially contributing to more effective T2DM management. The novelty and uniqueness of the proposed study are justified by below below-mentioned highlights

1. This design will predict and compare the CGM data from two dimensional perspectives. *i.e.,* time-series and regression to analyze dependent and independent factors affecting BG values.
2. The self-learning system improves the BG control by monitoring and comparing predicted results. Hence, diet improvement continues based on AI-solution advice.
3. The proposed design does not require traditional finger-pricks to record BG values, manual data entry by patients for observed readings, or any clinical intervention.
4. Most importantly, our design is not a treatment but prevention achieved by managing T2DM keeping the BG graphs smooth.

T2DM patients with this disease may achieve a lot by adopting a healthy lifestyle and sensible amendments in their diet intake, advised at individuals' predictive glucose levels. Later, observe and compare the outcome; obtained from the AI-based suggested diet plan and its effect on reducing glucose levels' peaks. In addition, this study concluded to overcome patients' limitations during clinical intervention. The blood glucose time-series data is exported through a continuous glucose monitor sensor inserted under the patient skin for fourteen days.

The flow of this article is done in a sequence, the next section discusses related work along with SIRD model, classification, and time-series analysis. After that, the section is about research methodology that covers dataset collection, dataset description, modeling technique, result, and discussion. The performance metrics are calculated to validate the fitted model for a given dataset. Later, the section is the next level of our research elaborating on AI-based dietary solutions derived from the results of prediction from the previous section. Finally, the conclusion and future work are discussed in the last section.

## Related work

Numerous studies have been conducted concerning machine learning, and it has now further long-drawn-out into the field of forecasting. The main aim of medical science is to learn and estimate disease patterns so that they can be cured and managed accordingly. In prospect, the predicted way might help patients about when they can resume normal lives. The accuracy of forecasting blood glucose would aid in diabetes treatment by enabling proactive treatment (*Reifman et al., 2007*). Besides applying different algorithms to forecast, time series data for predicting and managing chronic diseases, many researchers

have shown their work to develop better algorithms. Hence, predicting blood glucose levels using various statistical and machine learning methods (*Oviedo et al., 2017*). This has also inspired the Ohio type 1 diabetes mellitus (T1DM) challenge mentioned in *Marling & Bunescu (2020)*, where tasks were given to participants to forecast blood glucose results in individuals with type 1 diabetes mellitus (T1DM) accurately.

One such research was published in 2021 to forecast time series patterns four weeks ahead. The algorithm was applied on the previous year's COVID-19 data record, *i.e.,* from January 20, 2020, to May 21, 2020, collected in Indonesia (*Satrio et al., 2021*). There are many autoregressive algorithms. The adaptive moving average (ARIMA) model and FB-PROPHET are two common forecasting techniques, all discussed individually. These algorithms have been used in numerous fields, including the banking stock market (*Almasarweh & Alwadi, 2018*) and travel demand (*Petrevska, 2017*; *Petrevska, 2012*). Although PROPHET was only released five years ago, it can be regarded as a new approach because of its powerful model that is simple to use. The literature that people have used as approaches to predict, classify, and forecast different diseases is discussed in the sections below.

## SIRD model (classification and forecasting)

Much research employs the Scheme of the Susceptible-Infectious-Recovered/Death (SIRD) model to forecast disease. A few studies focused on SIR model to forecast Influenza outbreaks in Belgium and Hong Kong during the epidemic (*Yang et al., 2015*; *Miranda et al., 2019*). Similarly, the same model is implemented to track disease spread during the COVID-19 epidemic, (*Yang et al., 2020b*; *Yang et al., 2020a*). It makes following the epidemic's beginning, ending, and pattern variations possible. In one more study, the COVID-19 outbreak was predicted by *Anastassopoulou et al. (2020)* by using the SIRD model without considering factors that play an important role in finding out disease dynamics.

Which algorithm best fulfils the objective depends upon its performance; a similar study comparing sensitivity, specificity, and balanced-accuracy metrics of various classification algorithms is done (*Leo, Luhanga & Michael, 2019*). Further research to predict Parkinson's disease uses modified RNN and GRU in combination with SVM to classify input data mention (*Che et al., 2017*). Naive Bayes and the J48 Decision Tree were also used to construct infections due to MERS-CoV. The other algorithms may include multi-layer perceptron, LSTM with Bayesian ridge regression, multilayer perceptron, and gated recurrent unit (GRU). As a result of these comparisons, the most impressive display was made by GRU by feeding correct input (*Satrio et al., 2021*).

## Time series analysis

Time series data have 'Time' dependency, a sequence of data points collected over time. In China, a relevant study was done to predict the AIDS epidemic using search engine query as a dataset (*Nan & Gao, 2018*). The forecasting was done with the artificial neural network method. One other such high-accuracy forecasting of the zika virus was done in *Akhtar, Kraemer & Gardner (2019)*. This study is time series modeling with a nonlinear

**Table 2  Related work.**

| Authors | Study country | Target | ML model | Year |
|---|---|---|---|---|
| *Ravaut et al. (2021)* | Canada | New-onset T2DM with 5 years | Gradient boosting | 2021 |
| *Nomura et al. (2020)* | Japan | New-onset T2DM with 1 years | Gradient boosting | 2020 |
| *Huang et al. (2022)* | China | New-onset T2DM | Gradient boosting | 2020 |
| *Kopitar et al. (2020)* | Slovenia | Early detection of T2DM | Glmnet, LightGBM, XGBoost, Random forest | 2020 |
| *Simon, Zhang & Wang (2023)* | Korea | New-onset T2DM within 5 years | Logistic regression | 2019 |
| *Kistkins et al. (2023)* | – | Comparitive analysis T1DM | Logistic regression, ARIMA, LSTM | 2023 |

autoregressive exogenous model (NARX). In a study based on application to blood glucose prediction, deep residual time-series forecasting is implemented (*Rubin-Falcone, Fox & Wiens, 2020*). The researchers proposed a new architecture of learning to forecast in stages or blocks, replacing fully connected blocks with recurring neural networks and providing auxiliary supervision by adding additional losses. There are also many studies about Prophet forecasting and combining Prophet with ARIMA to mark a new model. One such example of combining both models is seen in *Ye (2019)*, in which weather forecasting is done against air quality, including pollutants in the air. As discussed previously about techniques to predict, forecast, and classify different diseases, we may say without any doubt that machine learning could be a tool to maximize predictive performance in contrast to traditional statistics models. Similarly, Table 2 summarizes the studies related to T2DM prediction done by various approaches of machine learning.

## METHODOLOGY

This proposed study aims to design a complete integrated system that aids diabetic patients in improving their lifestyle. In its development known approaches are used to train and test models and to get forecasted BG values. Later, validation is done and performance metrics are represented, all achieved with historical data obtained from CGM deployed at patients. Performance metrics are essential components of any machine learning task since they serve as standards for evaluation and serve to quantify advancement. They can be classified as either regression or classification methods. Our study is based on the Regression model which examines important metrics including mean absolute error (MAE), mean squared error (MSE), root mean squared error (RMSE), and R2 (R-squared) as the performance indicator. This study attempts to provide a thorough understanding of the principles underlying these metrics and their importance in assessing the model's performance. The research is divided into two layered roadmaps, the data and model layers respectively, as presented in Fig. 2. Traditionally, each layer is divided into three phases. The data layer, steps 1–3, implements preprocessing and data preparation. Step 4 of the model layer involves selecting model parameters, which can be done automatically or

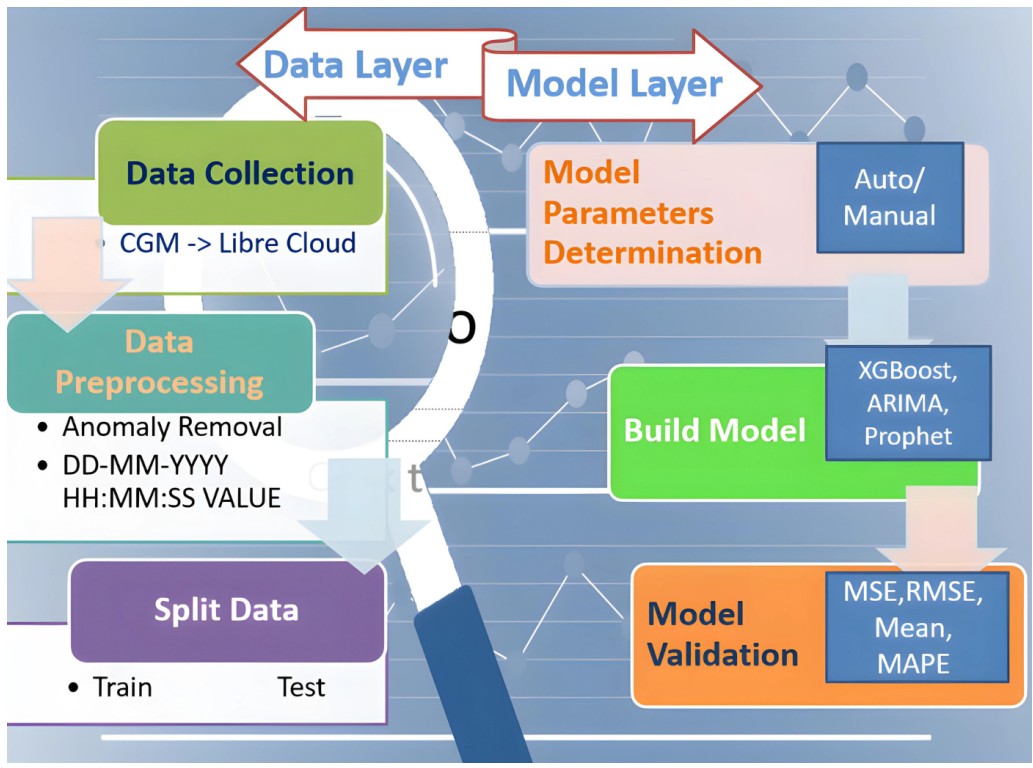

**Figure 2  Two-layered division of research roadmap.**

manually. The method of choice is determined by the library's ability to build the model. The following step 5 entails establishing the parameters specified in the previous step and training the model. At this point, all acts involving the model are called "model building". The model is validated in the final phase step 6. We mean "model validation" when we obtain a prediction and compare the parameter estimates to their actual values. We will use a set of indicators mentioned here to assess the model's quality. If the model metrics match the criteria, the model can be saved and ready to be employed to make a forecast. If the metrics are unsatisfactory, go back to step 4 and complete all further stages at the data layer. Five statistical metrics were used to estimate the model's accuracy: the correlation coefficient (R2), the mean squared error (MSE), the root mean squared error (RMSE), the mean absolute error (MAE), and the mean absolute percentage error (MAPE) (*Huang et al., 2022*). The correlation coefficient (R2) is used to calculate how well-anticipated values match actual values. MSE is defined as the average difference between actual and anticipated values, RMSE is the square root of MSE, and MAE is recommended as the average difference between real and predicted values. RMSE, as opposed to MAE, highlights the variance between data outliers.

## Dataset collection, description and preprocessing
### *Dataset collection*

In this study, we have used real-time BG data from the sensor. The data samples of blood glucose are taken from a continuous glucose monitor (CGM) from March 2021 to December 2021. CGM is a small sensor-based system that provides real-time glucose readings day and night for fifteen days against one device without conventional finger pricks, unlike traditional blood glucose meters (BGM) (*FreeStyle, 2023*). On the assessment days, the CGM data measured glycemic parameters such as mean glucose level, variability, time in the glycemic range, hyper and hypoglycemia, and duration. Sensor data is sent to the secure cloud-based server and is updated in an application specifically designed by the manufacturer to store and view results. The data is recorded, every time the patients scan the sensor with a smartphone (with NLC and manufacturer-provided app) or by its reader device. The monitoring system stores data in the database following preprocessing, provided that the patient scans the sensor within eight hours, seven days a week. This monitoring system can provide data for any period of interest in the format shown in Fig. 3.

### *Description and preprocessing*

The file obtained from the CGM device consists of five columns of primary information, including the device name with its serial number, timestamp, record type, and glucose in mmol/L. The original CSV file was then further updated by removing NaN (Not a Number) records. The total number of 1,578 NaN fields that contain no glucose number were removed before proceeding further into the time series algorithm.

The ML method needs diverse input data to develop results and predict patterns daily, monthly, and yearly. The amount of input features and size of the dataset significantly impact the model's output (*Ahmad et al., 2021*). A sensor attached to the skin and a plastic needle placed in a muscle is used to quantify the amount of glucose in the interstitial fluid. The handheld reader reads the sensor readings or compatible smartphone through a wireless receiver. The CGM sensor also has Bluetooth connectivity, allowing it to alert users with a sound or vibration if their BG is high or low in real-time without scanning. The desired results are independent of other constraints such as patient age, gender, BMI, and activity pattern. The CGM used in our study claims an accuracy of 9%+/- for the real-time monitor sensed results.

The model generally required comparable input parameters for each diabetic patient to yield the required output. The descriptive statistics for each input variable are given in Table 3. Here, descriptive statistics reflect a collection of brief, scientific measurements that provide an outcome, such as the entire population or its subgroup. This research used data samples of continuous glucose readings in multiple batches of fifteen days in a month to run the ML model. This study focused on three fields among various attributes, recorded for patients with CGM deployed. The data was obtained based on areas with each area's date, time, and real-time glucose readings. The dataset contained a total of 10,160 glucose records after cleaning data obtained from the CGM device, Table 4 shows the head and tail

| Glucose Data | | Generated on | 30-03-2023 18:21 UTC | Generated by | Mansoor Bin Saeed |
|---|---|---|---|---|---|
| 0 | Device | Serial Number | Device Timestamp | Record Type | Historic Glucose mmol/L |
| 1 | FreeStyle LibreLink | 973fbf26-2682-45ef-b004-37919b9f4586 | 4/28/2021 13:01 | 0 | 15 |
| 2 | FreeStyle LibreLink | 973fbf26-2682-45ef-b004-37919b9f4586 | 4/28/2021 13:16 | 0 | 14.8 |
| 3 | FreeStyle LibreLink | 973fbf26-2682-45ef-b004-37919b9f4586 | 4/28/2021 13:31 | 0 | 14.1 |
| 4 | FreeStyle LibreLink | 973fbf26-2682-45ef-b004-37919b9f4586 | 4/28/2021 13:46 | 0 | 13.9 |

**Figure 3** Original dataset slice as received from CGM device.

**Table 3** Dataset description and demographics.

| Parameter | Values/Glucose monitor |
|---|---|
| Patient age | 40/45 |
| Patient gender | Male |
| Patient location | United Kingdom |
| Start date | 28-April-2021 |
| End date | 24-December-2021 |
| Time interval | 15-Minutes |
| Count | 10160 |
| Mean | 8.295 |
| Std | 2.584 |
| Min | 3.4 |
| 25% | 6.4 |
| 50% | 7.8 |
| 75% | 9.9 |
| Max | 19.9 |

records of the original dataset. It contains the date and time stamps along with the glucose value.

In the data preprocessing phase, columns are created for year, month, day_of_week, time, and day_name respectively originally extracted from datetime stamp information as shown in Table 5. Later, the correlation among columns is checked and features with high correlation are eliminated from the data frame to reduce dimensionality. For now, the year is dropped as it is highly correlated with the month, but in the future, it can play an important dimensionality and can be added later for data expanding on multiple years.

## Graphical representations and feature engineering

The percentile graph in Fig. 4 shows showing percentage of BG values for different ranges in our dataset for 24 hrs. From the data analysis, it is observed that only 0.1% of BG values are below the range of 7 (mmol/L), whereas 0.9% BG values in the data set compromise range between 10–13 (mmol/L). From these observations, we can conclude the occurrence of hyperglycemic episodes is way larger than hypoglycemic. The initial dataset is split into train and test datasets with a ratio of 75% and 25% respectively. This splitting is done by assigning a test data size 0.25. Figure 5 shows the periodicity of the predicting process, with

**Table 4   Cleaned dataset BG values. dataset head and tail.**

| Record Number | Time | Glucose |
|---|---|---|
| 0 | 4/28/2021 13:01 | 15.0 |
| 1 | 4/28/2021 13:16 | 14.8 |
| 2 | 4/28/2021 13:31 | 14.1 |
| 3 | 4/28/2021 13:46 | 13.9 |
| 4 | 4/28/2021 14:00 | 13.9 |
| .. | ................ | .... |
| .. | ................ | .... |
| .. | ................ | .... |
| 10155 | 9/24/2022 11:50 | 9.9 |
| 10156 | 9/24/2022 12:05 | 8.7 |
| 10157 | 9/24/2022 12:20 | 8.7 |
| 10158 | 9/24/2022 12:35 | 8.9 |
| 10159 | 9/24/2022 12:50 | 8.6 |

**Table 5   Data analysis: BG values w.r.t year, month, day of the week, name of week and time respectively.**

| Index | Time | Glucose | Year | Month | Day of week | Time | Day name |
|---|---|---|---|---|---|---|---|
| 0 | 2021-04-28 13:01:00 | 15.0 | 2021 | 4 | 2 | 13:01:00 | Wednesday |
| 1 | 2021-04-28 13:16:00 | 14.8 | 2021 | 4 | 2 | 13:16:00 | Wednesday |
| 2 | 2021-04-28 13:31:00 | 14.1 | 2021 | 4 | 2 | 13:31:00 | Wednesday |
| 3 | 2021-04-28 13:46:00 | 13.9 | 2021 | 4 | 2 | 13:46:00 | Wednesday |
| 4 | 2021-04-28 14:00:00 | 13.9 | 2021 | 4 | 2 | 14:00:00 | Wednesday |

the seasoning period taken here being eight months. The blue color marks show test data which is used to test the model, whereas, the orange marks depict train data. Forecasting of BG values is done by using this train data set, by first training our predictive models, discussed in a later section.

The asymmetric measurement ranges of hypoglycemic index 2.2–3.9 mmol/L (40–70 mg/dL) and the hyperglycemic index 10 mmol/L–22 mmol/L(180–400+ mg/dL), the former is narrower than later, may result in misleading information with the blood glucose standard deviation value (STD). The distribution of glucose values is highly skewed, so the STD can be influenced predominantly by hyperglycemic excursions and not sensitive to hypoglycemia. For this reason, the interquartile range (IQR) is a more suitable measurement for non-symmetric distributions. IQR from the dataset is calculated before plotting the graph as shows in Fig. 6. Here the BG value range 2.2 mmol/L (the minimum detectable by the CGM) to 11.1 mmol/L is illustrated, with points above then 11.1mmol/L identified as outliers which are calculated by the IQR.

This box graph is in Fig. 7 window shows everyday blood glucose values for a whole week, in addition to minimum and maximum values for that day. The information depicted in this graph shows intriguing observations. The points outside of the top "whisker" are

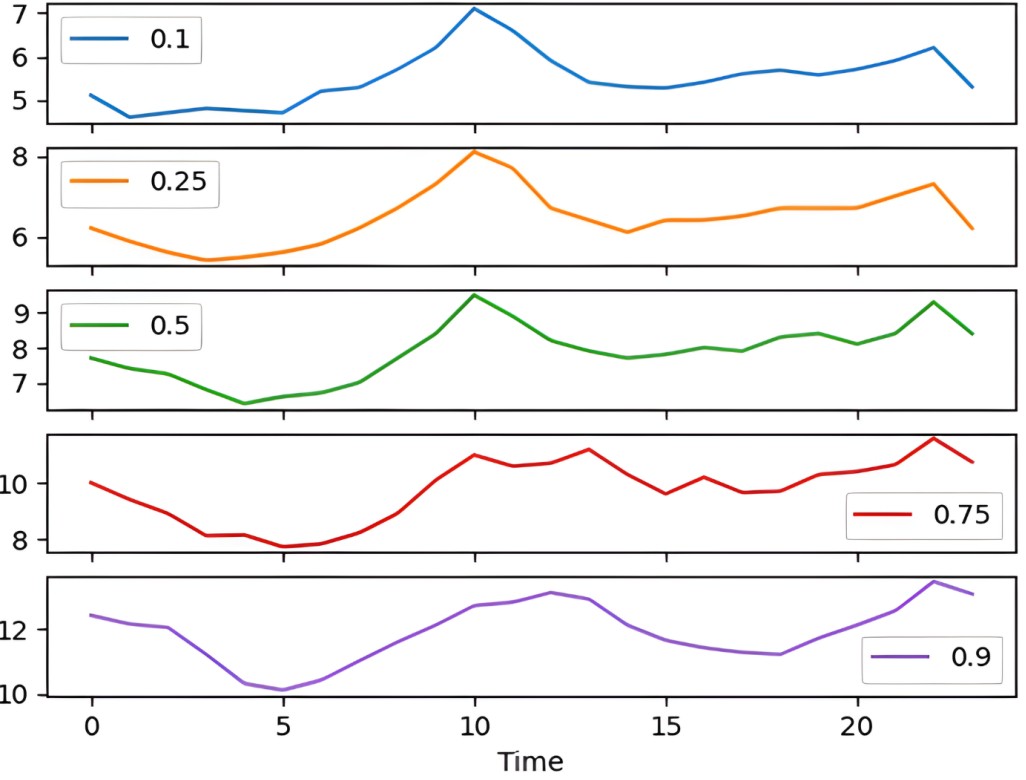

**Figure 4** Percentile distribution of BG range.

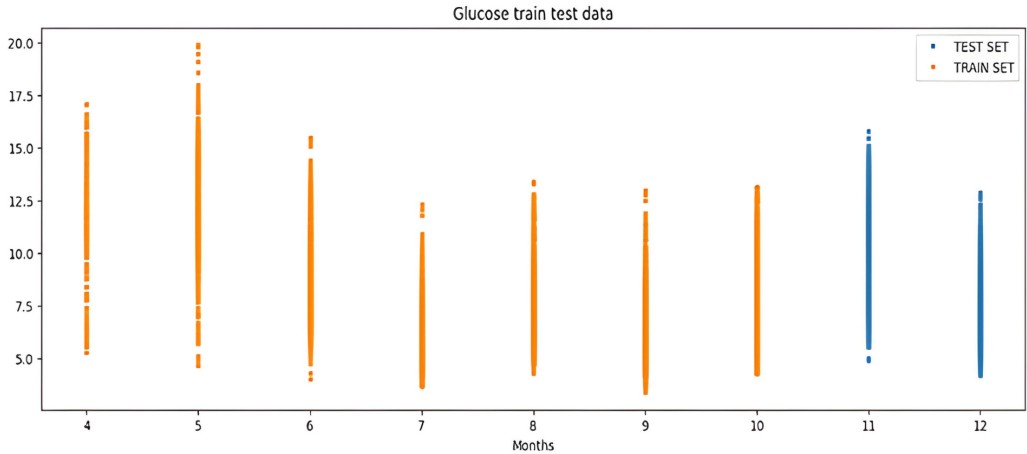

**Figure 5** Periodicity of the predicting process.

outliers, the observations that are numerically distant from the rest of the data. The IQR on Sundays and Mondays seemed to be quite small, showing high BG level excursion.

Effective feature engineering is vital for enhancing model performance, facilitating more precise predictions, and ensuring the model's robust generalization to unseen future data

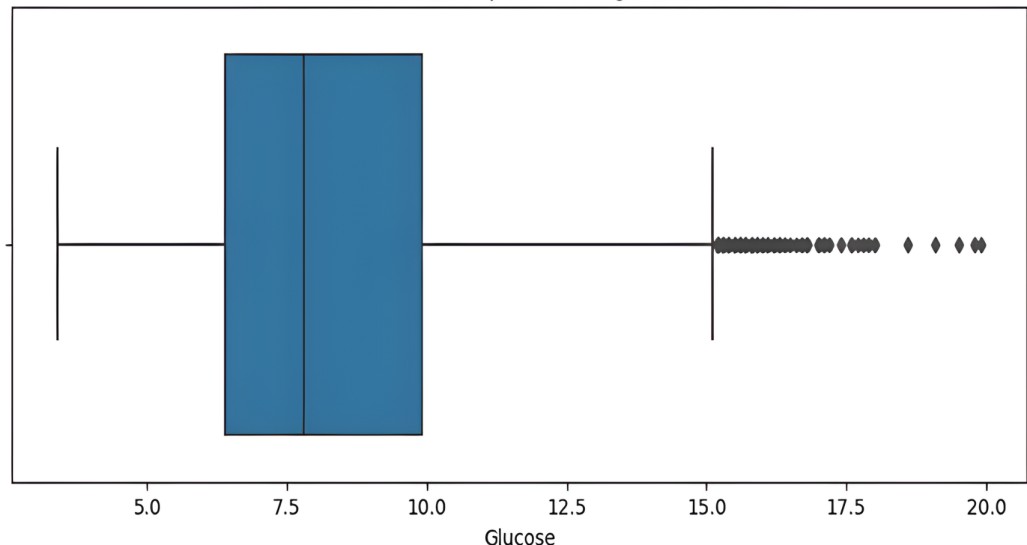

**Figure 6** Interquartile range of BG values.

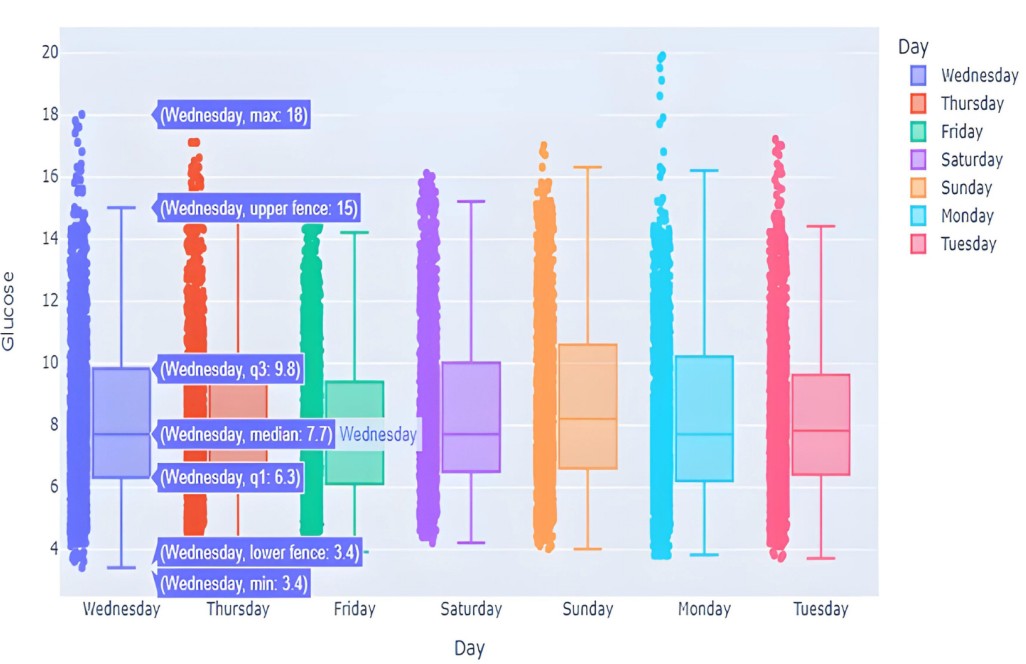

**Figure 7** BG distribution over week with outliers.

**Table 6  Feature engineering on the CGM dataset.**

| Feature | Formula | Meaning |
|---|---|---|
| CGM Lag | CGM (t 1) = CGM(t) * Shift (1) | One step lag of current CGM value corresponding to time axis |
| CGM Differential | 1st order: dCGM(t)=CGM(t) CGM(t-1), 2nd order: D2CGM(t)=dCGM(t) - dCGM(t-1) | The differences of order N between successive CGM values. |
| Rolling Range | RollingMIN(t, N)= min(CGM(t0, CGM(t-1)), …. , CGM(t-N+1) RollingMAX(t, N)= max(CGM(t0, CGM(t-1)), …. , CGM(t-N+1) | The current value's position within the rolling range is indicated by the Range Oscillator, where a value of 1 signifies its placement at the top, 0 at the bottom, and a value in between denotes its relative position. |
| Glycemia Class | +1 (hyper) if CGM(t) >180mg/dL(10mmol/L), GC(t)= 0(norm) if 70mg/dL (3.9mmol/L) >CGM(t) <180 (10mmol/L) -1(hypo) if CGM(t) <70 mg/dL (3.9mmol/L) | Our goal is essentially to predict whether it will be hypoglycemia, hyperglycemia, or a normal state. |

points. Consequently, it formed the cornerstone of our methodology. The features have been categorized into the groups outlined in the Table 6.

## Development of model

Once the data has been collected, analyzed, and preprocessed. The next step is to build a model and predict the forecast. Figure 8 depicts the sequential approach used in this study to achieve the required objectives. The forecasted output will be categorized $+1$, $0$, or $-1$ according to glycemic class as mentioned in the previous section. The weights assigned to different output ranges will give a quick thought to patients in choosing the next course of action in adjusting their diet.

The model is built and implemented in Python, using the Google Colab environment to achieve the desired results. Google Colab is a cloud-based platform with a Jupyter Notebook interface, allowing seamless integration with Python libraries for data science and machine learning projects. This cloud-based approach removes the need for powerful local machines calculated on Google's servers. Additionally, Google Collab integrates with Google Drive to facilitate storing and retrieving datasets, models, and notebooks. Implementing the model in Google Collab increases accessibility, scalability, and performance, making it desirable for engineering applications in data science and machine learning (*Almufarreh, Noaman & Saeed, 2023*).

In this study we compared forecasted BG values obtained from the development of three different models; based on XGBoost, SARIMA and Prophet algorithms. Following the training phase, forecasts are generated for the testing data, and performance metrics including MAE, MSE, and RMSE are computed later. All of the models in this study are trained on train data to forecast 2-month BG values.

## Time series forecasting algorithms
### XGBoost

Extreme gradient boosting (XGBoost) is used to develop a regression model to predict BG values. This regression algorithm is known for its efficiency in handling structured data and uses a cluster learning approach. This can be achieved by combining multiple weak learners, typically decision trees, and predictions to form a complex model. This is a

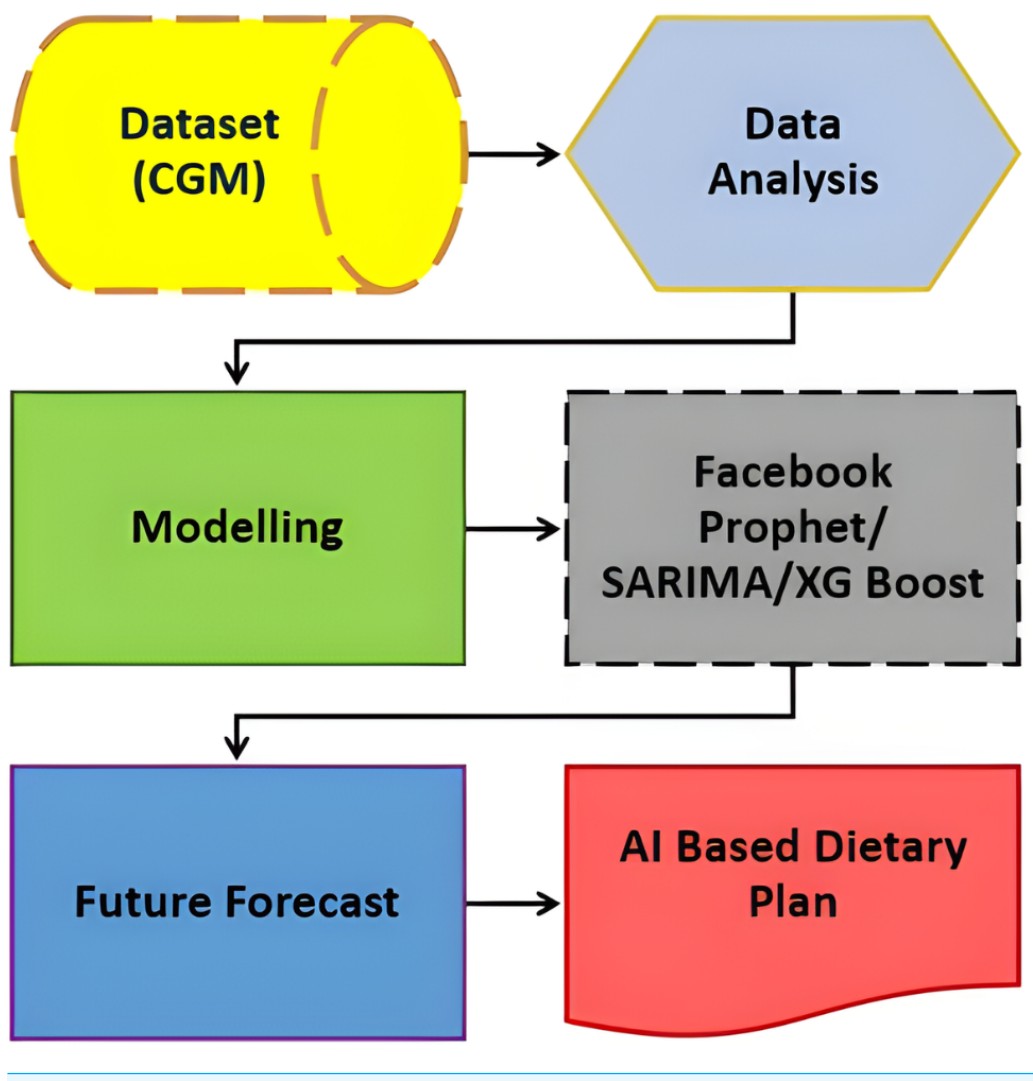

**Figure 8** Sequential approach.

distributed pot grow tool designed to be efficient, scalable and portable. Gradient boosting framework is used to develop machine learning algorithms. XGBoost provides parallel tree enhancement (GBDT, also known as GBM) that addresses many data science issues quickly and accurately. The same algorithm runs in large distributed environments like SGE, MPI, and Hadoop, and can handle problems with billions of instances.

The predicted BG values from the supervised XGboost regression model are shown in Fig. 9. The train and test data sets confounded 75% and 25% of the original data, respectively. The model is built using the XGBRegressor function from its library. The program has a very large number of parameters, which are configured to create the required model. Once the model is trained, the prediction is made two months later on the test data. In addition, the XGBoost regression model includes regularization methods to prevent overfitting. However, special attention was also paid to tuning hyperparameters,
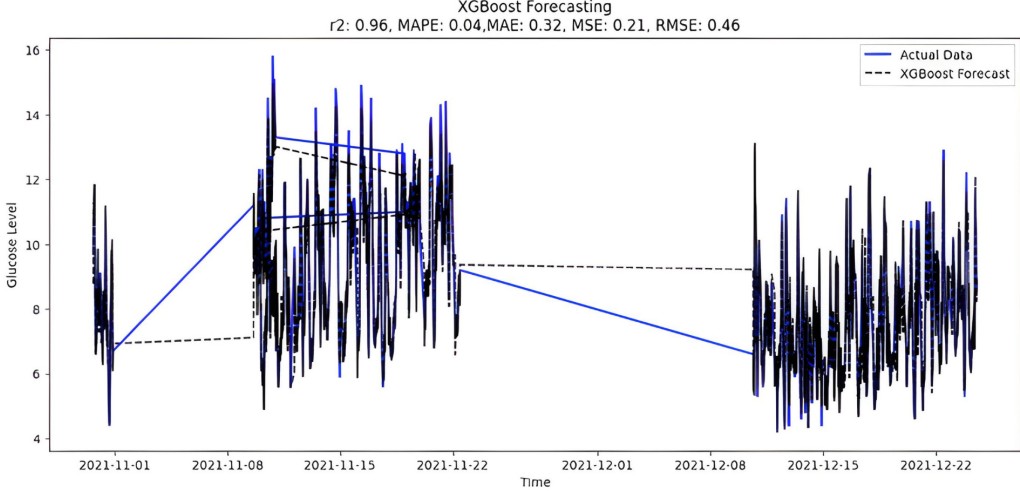

**Figure 9** XGBoost forecasting.

changing parameters such as learning rate and tree depth to achieve optimal performance. Below are the XGBoost regression model parameters:

1. max_depth = 6 (is used to define maximum tree depth for base learners)
2. learning_rate = 0.05 (boosting learning rate, the smaller the better
3. n_estimators = 500—number of gradients boosted trees (boosting rounds)
4. gamma = 0—minimum loss reduction required to make a further partition on a leaf node of the tree.
5. Learning_rate = 0.30000

The features, Lag, and Rolling obtained from feature engineering are also utilized in our model. Lags for the last three time points of glucose values are taken to convert time series forecasting as a supervised machine learning problem. Similarly, the rolling window statistic is also added including mean, median, and std deviation over a fixed window size statistics of eight months of data.

### SARIMA

After closely examining the various features of the BG price data set, we categorized them as volatile and seasonal; Therefore, in developing our model, we used seasonal ARIMA instead of ARIMA. The annual seasonal auto-regressive integrated moving average is an extension of ARIMA that supports time series data with seasonal components. It includes auto-regression, differentiation, and moving average components to capture temporal patterns. In the initial step, the model is configured using the SARIMA class from the stats models library. After training the historical data, the SARIMA model is suitable for predicting periods up to two months in the future. This model handles non-stationary data through automatic differentiation. However, choosing the appropriate order (p, d, q) is important and may require a thorough examination of autocorrelation and partial autocorrelation functions.

Before explaining the mathematical expression for seasonal ARIMA, we first considered coefficients for ARIMA, with an order of p, d, and q respectively for the stationary part of data. These seasonal differences then allow dealing with seasonality that varies in amplitude from every eight months in our model, *i.e.,* modeling the multiplicative seasonality *via* ARIMA by making the seasonality itself stable. Hence, our model can be defined with wholesome seasonal ARIMA(P,D,Q)m features, comparable to the ARIMA(p,d,q);

$$yt(1 - Bm)D(1 - \phi m, 1Bm - \cdots - \phi m, PBp.m.) = \varepsilon t(1 + \theta m, 1Bm + \cdots + \theta m, QBQm)$$

if,

$$\Delta D(Bm) = (1 - Bm)D$$
$$\varphi P(Bm) = 1 - \phi m, 1Bm - \cdots - \phi m, PBp \cdot m.$$
$$\vartheta Q(Bm) = 1 + \theta m, 1Bm + \cdots + \theta m, QBQm.$$

The seasonal ARIMA(p,d,q)(P,D,Q)m model can also be succinctly expressed after regrouping the backward shift operator.

$$yt\Delta D(Bm)\varphi P(Bm)\Delta d(B)\varphi p(B) = \epsilon t\vartheta Q(Bm)\vartheta_q(B)$$

The parameters are optimistically configured first for a non-seasonal part, *i.e.,* order (p, d, and q) with later seasonal data (P, D, Q) m, respectively. The augmented Dickey-Fuller test is done to check the stationary nature within the dataset. First and second-order differentials with respective plots are generated by feature engineering mentioned earlier. The coefficient 'P', defining auto-regression is calculated from the partial auto-correlation (PACF) graph, noticing several lags crossing the upper and lower bound of a significant area within it. Whereas, the coefficient 'Q' tells about the moving average that is dependent upon the auto-correlation (ACF) graph. Lastly, the Seasonal parameter 'm' is set to 8 in our model due to seasonality stretching out to eight months in the under-study dataset. As data is seasonal and cyclic for every eight months, PACF and ACF graphs are calculated from 8th differential function taken on BG values. It is important to observe that the symbol 'd' denotes the integration order of the seasonal process. It is equivalent to several transformations done when the seasonal time-series data behave stationary. It is fixed for the value that made the ADF hypothesis correct, where, $p$-value $< 0.05$ for time-series stationary and $p$-value $> 0.05$ for time-series non-stationary. The ADfuller test and implementation of SARIMA is done with the help of the stats models library.

The results of the augmented Dickey-Fuller test are mentioned below;
$p$-value original: 7.294737315993314e-08
number of Lags:96
ADF statistics: $-6.157850$
Critical Values:
1%: $-3.431$
5%: $-2.862$

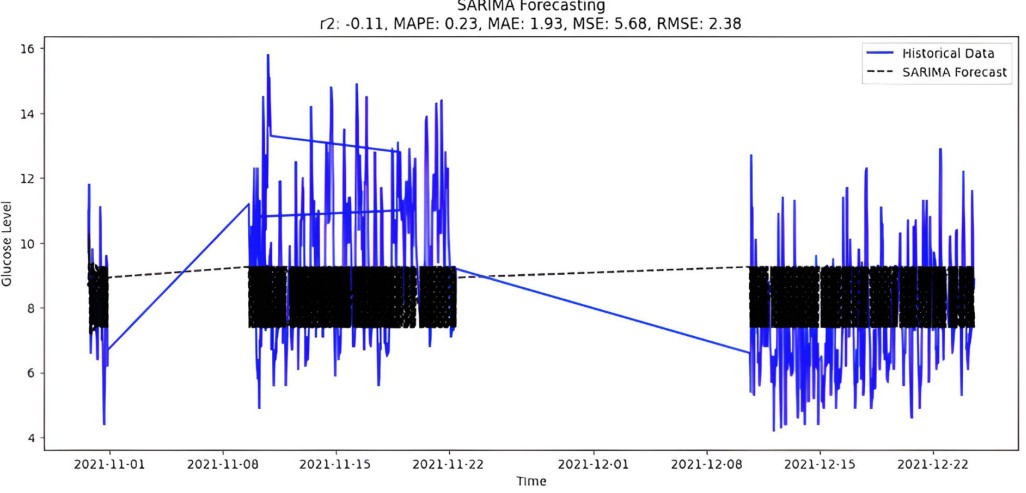

**Figure 10  SARIMA forecasting.**

10%: −2.567

The selection of parameters(p,d,q)(P,D,Q) to fit the model is (4,0,2)(0,1,0,8), which is the best possible fit for study data. It is also possible to select parameters automatically by using 'auto_arima' function and the forecast presented in Fig. 10.

### *Prophet*

Prophet, a timeline prediction tool developed by Facebook, is used in the code to predict glucose levels. Accredited for flexibility and ease of use, Prophet uses additive modeling to decompose timeline data into features such as trends, seasonal features, and holidays. Prophet's flexibility is evident in its ability to handle lost data and outbound traffic, allowing users to add custom seasonality and holidays. The model also automatically detects the change points, which can be important for capturing abrupt changes in the time series. Users should consider including appropriate seasonal and holiday effects to increase forecast accuracy.

Prophet also pointed out that internal performance is difficult to define when dealing with explicit temporal data models. Algorithms have applications in a variety of industries, including finance, social media, and healthcare. Notably, its specialization provided accurate sales forecasting for the trading industry, indicating its versatility. The Prophet proves invaluable in capturing and anticipating trends in various forms. Seasonal results, represented by outliers in weekly, monthly, and annual cycles, are best achieved using algorithms. In summary, Prophet is emerging as a powerful and versatile tool for time series forecasting.

$$y(t) : g(t) + s(t) + h(t) + E * t$$

y(t) - Reversal model to add g(t) - trends (t) - season $\varepsilon * t$ - Error

The forecast as depicted in Fig. 11 is achieved as per the equations above, where y(t) is the target value that is to be predicted, g(t) is the trend term, and s(t) is the season term, which will depend upon the data periodicity, *i.e.*, either as weekly, yearly or intraday. It

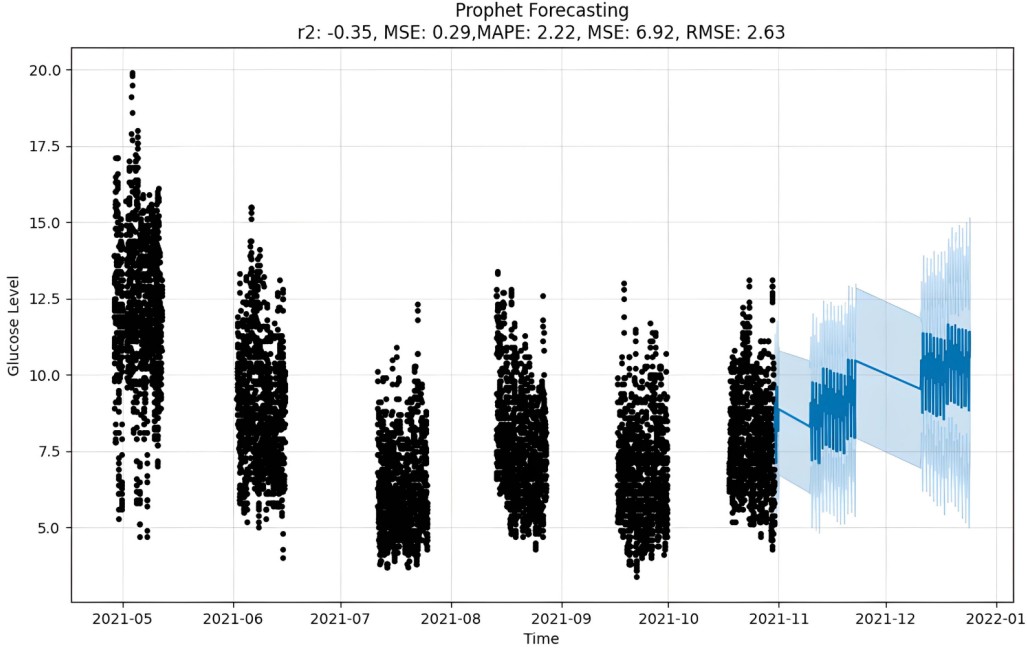

**Figure 11  Prophet forecasting.**

also allows for custom holidays and is mentioned by the variable h(t). At the same time, the E is an error term assumed to be normally distributed over variables (*Saxena, Indervati & Rathi, 2022*). For predicting time series, numerous methods can be applied to make precise predictions. One such option is MAPE that predicted by calculating mean absolute percentage error. This gives us the tools we need to predict timetables accurately using natural parameters, and there is a setup for capturing the perspective of the season and holiday heritage. On the other hand, Prophet tries to make comparisons between various direct and indirect time activities, like objects. The exponential smoothness employs the same tactic of modeling the season as an add-on. This library is so significant that it can stand alone in features related to data and the seasons. However, Prophet has some restrictions, such as the assumption that input columns with the names "ds" and "y", where "ds" stands for date and "y", the target variable, are present. In this case, a trend may vary between upper and lower BG peaks *i.e.,* hyperglycemic and hypoglycemic values.

## RESULTS AND DISCUSSION

Upon scrutinizing the error rates associated with each forecasting algorithm, it becomes evident that XGBoost outperforms both SARIMA and Prophet in terms of predictive accuracy for the provided time series data. XGBoost demonstrates remarkably low errors, Fig. 12 and Table 7 showcasing a mean absolute error (MAE) of 0.32, mean squared error (MSE) of 0.31, and root mean squared error (RMSE) of 0.46. Additionally, the model attains a high coefficient of determination ($R^2$) of 0.96 and a low mean absolute percentage error (MAPE) of 0.04, indicating an excellent fit to the data and minimal percentage

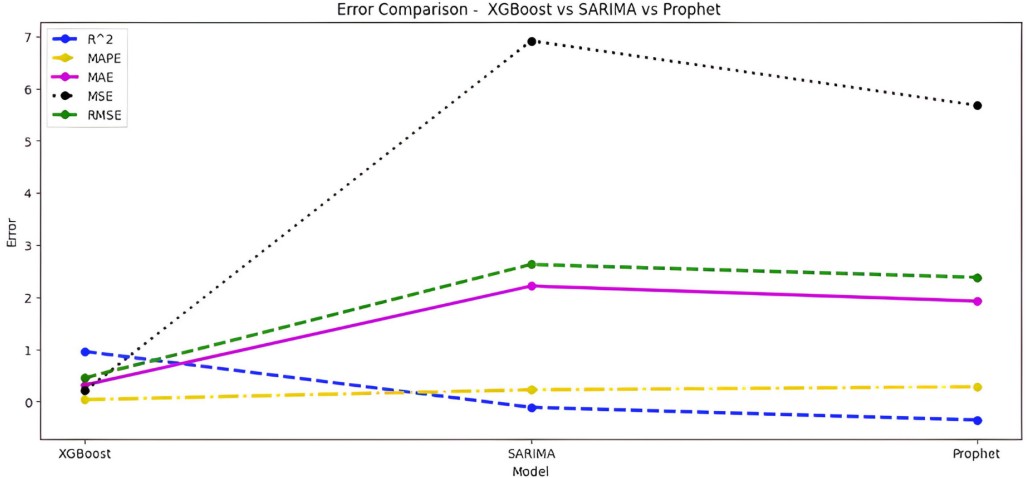

**Figure 12  Comparative analysis of MAE, MSE and RMSE for models.**

**Table 7  Error values.**

| Algorithm-Error | XGBoost | SARIMA | Prophet |
|---|---|---|---|
| $\mathbb{R}^2$ | 0.96 | −0.11 | −0.35 |
| MAPE | 0.04 | 0.23 | 0.29 |
| MAE | 0.32 | 1.95 | 2.22 |
| MSE | 0.31 | 5.61 | 6.92 |
| RMSE | 0.46 | 2.37 | 2.63 |

deviation between predicted and actual values. It should be noted, ($\mathbb{R}^2$) with value higher than 0.9 is strongly acceptable, in this study, results of XGBoost showing excellent model behaviour with ($\mathbb{R}^2$) approaching to '1'. This factor is directly dependent on type and size of dataset for training model but shows insignificant behaviour with large sample size. Conversely, SARIMA displays moderate performance but registers higher errors across all metrics, including an $\mathbb{R}^2$ of −0.11, MAPE of 0.23, and RMSE of 2.37. Meanwhile, Prophet yields the least accurate predictions among the three models, evidenced by its relatively higher MAE (2.22), RMSE (2.63), $\mathbb{R}^2$ (−0.35), and MAPE (0.29) values. It should be noted that negative nature of ($\mathbb{R}^2$) is signaling model SARIMA and model FBProphet are not good fit with CGM dataset.

It is worth highlighting that, unlike the implementation of the three models, the XGBoost forecasted result exhibits a pattern comparable to the typical GC (t). Figure 13 shows a comparative analysis of forecasted BG outcome from each model in contrast to the observed BG values of this study. In this study, we have successfully done forecasting of BG values with the help of data acquired by CGM. From related studies, it has been observed that numerous factors are required to achieve similar goals, hence overburdening patients and their caretakers. In one such similar study (*Dinh et al., 2019*), diabetes prediction is done by making use of more than 120 attributes, acquiring data at this significant score

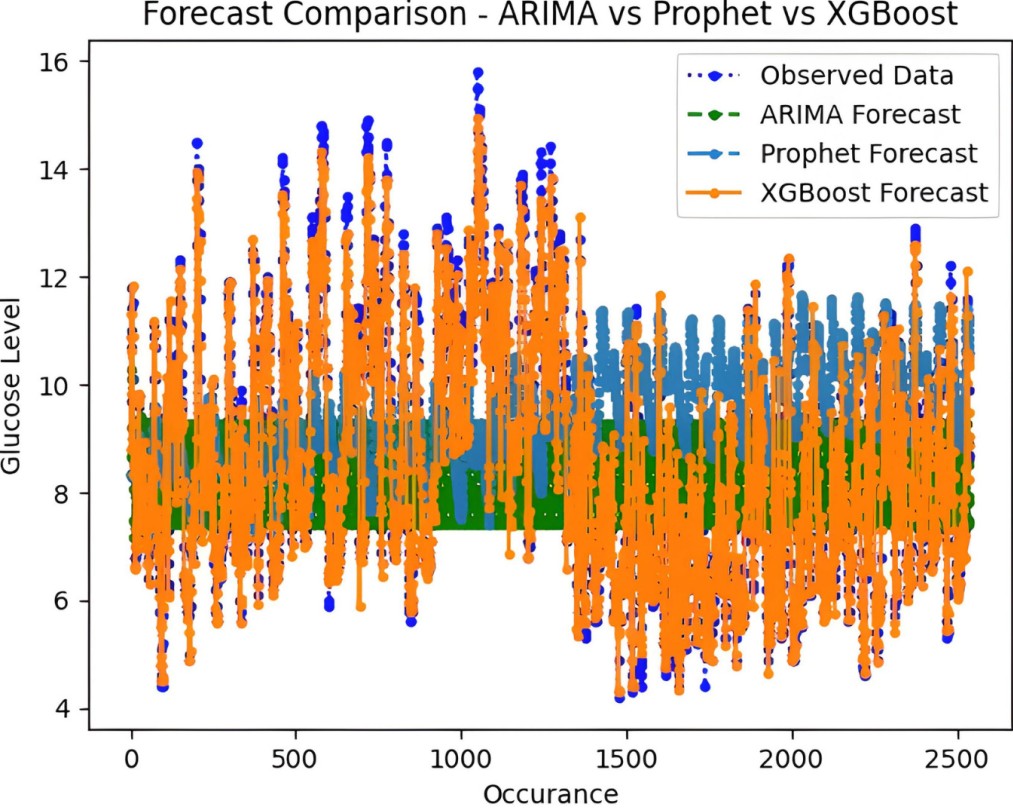

**Figure 13** Comparative analysis of forecasted *vs* observed BG values.

is almost impossible. In our study, T2DM management, only two parameters are crucial; First, BG values concerning date and time obtained by sensors. Second, the photograph of meals included in the patient's diet. Later, results were also compared based on the laboratory test for HBA1c that was done twice, first at the beginning and the other after three months post-intervention of our study. These comparisons for the person under observation showed decreased values, signaling controlled glucose peaks.

## Remarks

The results presented in the above section underscore XGBoost's superior suitability for the specific forecasting task at hand, emphasizing the significance of algorithm performance metrics in model selection for time series prediction. The XGBoost model is presumed to account for several anomalies contributing to the cumulative MAE. Such discrepancies arise from inaccuracies, unpredictabilities, or errors in the data obtained from the sensor instrument. To address this issue, it is imperative to broaden the research scope, obtaining another true value of AI meticulously from the sensor instrument. This methodological approach aims to address the identified shortcomings, thereby enhancing the overall accuracy and reliability of the model through a comprehensive and precise understanding of the underlying data dynamics.

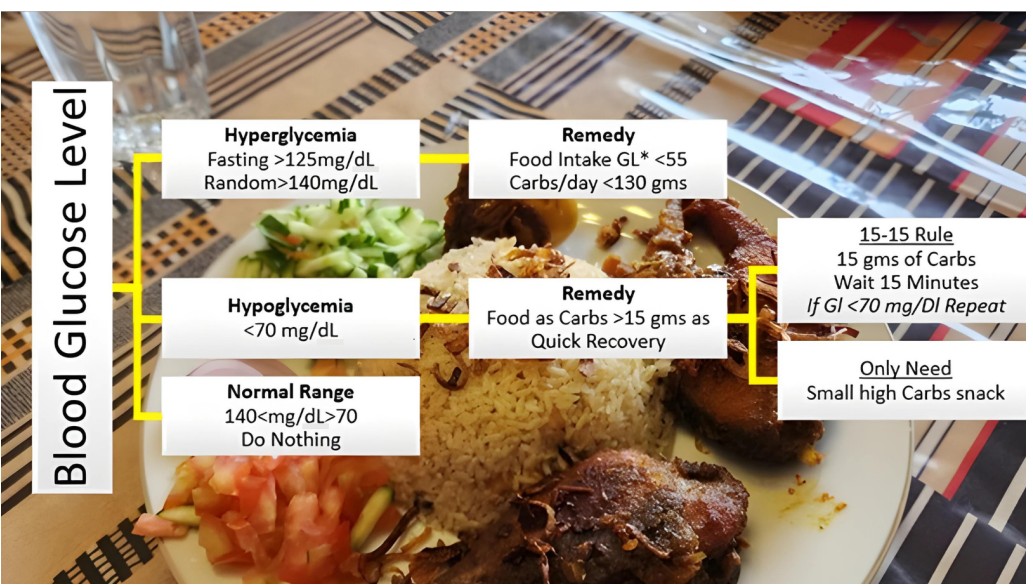

**Figure 14** **Food intake categories according to BG levels.** The Glycemic Index (GI) is a tool that measures how carbohydrates affect blood glucose levels. GI Low: 55 or less, Mid: 56-69, High: 70+ (*Glycemic Index Foundation, 2023*).

## Proposed AI-Diet adviser

The models, XGBoost, SARIMA and Prophet as expounded in the previous section, are deployed to anticipate the BG levels in the T2DM patients. To achieve the required intent, this study has further extended by proposing the integration of the forecasted results with AI-based dietary solutions, already available commercially. This will help guide the T2DM patient with the best possible dietary alternatives concerning real-time BG values. Initially, the food intake of T2DM patients is classified into different bins, categorically dividing according to glucose index as explained in Fig. 14. Predominantly, the dietary changes for diabetic people mostly involve guidelines to help keep blood glucose, blood pressure, and cholesterol levels balanced. For a healthy alternative, T2DM patients focus on a diet that is rich in essential nutrients considered as a complete meal; inclusive of fruits, vegetables, whole grains, and lean protein. On the contrary, contemporaneous with limiting the food intake having high glucose index, salt, saturated and trans fat. In this study, food intake options were limited and set to elementary, readily available items at home. The three categories according to three possible glucose ranges are introduced for different food types of T2DM patients; hyperglycemic, normal, and hypoglycemic.

The forecasted results obtained from different models, preferably XGBoost are then assigned weights according to glycemic class GC(t) introduced earlier in feature engineering. This will help T2DM patients by responding within a short time to determine the next course of action. Results within the normal range mean the patient's diet should be continued without any change. In this scenario, the system will stay in sleep mode, that is, do nothing. On the contrary, results with out-of-range BG levels mean either a hyperglycemic or hypoglycemic episode is underway, and the patient needs assistance for

intake food management. AI-based solution intervention at this point will assist the patient with diet change if required.

The proposed AI-based option in our study is "FoodLens" (by DoingLab Co., Ltd., Seoul, South Korea) due to its advisability and convenient accessibility. It can recognize numerous varieties of food through a single picture. The patient needs to provide information about his meal through photographs. This solution will identify and highlight seven major food nutrients among 40 different kinds, in a single go. FoodLens is a blend of convolution neural networks and deep learning algorithm approaches to classify food. It claims to achieve a high recognition rate a certain of 86.6% in food classification, also approved by Korea Information Security Technology (KOIST) (*Park et al., 2020*). Once the T2DM patient updates his meal photograph, this dietary solution will provide information about the total calories presumed through the image, whilst displaying reckoned food groups, including the amount of carbs, Proteins, Fats, and Fiber respectively. Therefore, this study in which BG values are forecasted based on a real-time continuous glucose monitoring system, got well to incorporate data through an AI-based nutrition solution.

## LIMITATIONS OF WORK

In this study, we have faced some limitations. The most important limitation was access to CGM sensor dataset. In most scenarios, T2DM patients usually use CGM devices on their own in homes, without the intervention of medical staff. Therefore, the record of such patients was not available to paramedic staff in hospitals. Although the target was a large community we managed to collect CGM BG value records from eight patients only. With large data, models can be trained, evaluated, and optimized in a better way. The performance accuracy of forecasting primarily depends on the availability of large amounts of data (*Kabiru et al., 2014*). A larger dataset means results will be more precise with less error rate. The one big challenge and a serious concern may arise due to lagging CGM results from a device with actual time. During our study, it has been observed that the actual readings obtained from CGM devices attached to patients under observation, were delayed by $+/-30$ min. This lag as a whole may add up in forecasted glucose results, affecting accuracy.

## CONCLUSION AND FUTURE WORKS

Patients frequently do not use healthcare smartphone applications for self-management of diabetes despite their availability due to various drawbacks, such as the inability to record dietary information by text search and the inconvenience of self-glucose monitoring by home glucometer. In this study, we effectively overcame by forecasting BG values based on data obtained from CGM and integrating with commercially available AI diet advisors. Hence timely adjusting T2DM patients' diet to reduce occurrence of hypoglycaemia and hyperglycemia episodes.

This study considered different methods to forecast time-series glucose data, which is done with XGBoost, SARIMA, and Prophet algorithms representing trends and seasonality. It is also concluded that the XGBoost model is best for such type of seasonal data with

comparison to SARIMA and Prophet by comparing performance metrics. As the sensor data is in the continuity of 15 days with a gap of 15 mins approximately between two consecutive readings, not every algorithm can perform well. The SARIMA method has a high error rate and its application in systems with non-stationary data having seasonal effects is complex to implement.

In the future, our target is to build a unique model by combining time-series and regression forecasting techniques to predict only hyperglycemic and hypoglycemic data. And study the effect of individuals' lifestyles effecting the abrupt fluctuations in BG values. For our current study targeting type 2 diabetes mellitus (T2DM), the prediction model will be optimized by tuning critical parameters in building the model. This will involve the collection of colossal data and upgrade to efficient computational resources, greatly enhancing the overall precision and accuracy. While collecting data, it would be more interesting to focus on the patients with T2DM who regularly take their medicines and follow healthy, consistent routines to benefit at maximum even with slight suggested changes in their patterns.

### Funding
This work was funded by the Deanship of Scientific Research, Vice Presidency for Graduate Studies and Scientific Research, King Faisal University, Saudi Arabia [INST. 041], through the KFU Annual Institutional Funding Program. The funders had no role in study design, data collection and analysis, decision to publish, or preparation of the manuscript.

### Grant Disclosures
The following grant information was disclosed by the authors:
The Deanship of Scientific Research, Vice Presidency for Graduate Studies and Scientific Research, King Faisal University, Saudi Arabia, through the KFU Annual Institutional Funding Program: INST. 041.

### Competing Interests
The authors declare there are no competing interests.

### Author Contributions
- Madiha Anjum conceived and designed the experiments, performed the experiments, analyzed the data, performed the computation work, prepared figures and/or tables, authored or reviewed drafts of the article, and approved the final draft.
- Raazia Saher conceived and designed the experiments, prepared figures and/or tables, and approved the final draft.
- Muhammad Noman Saeed performed the experiments, analyzed the data, authored or reviewed drafts of the article, and approved the final draft.

### Data Availability
The dataset and description attached with the submission.

## Supplemental Information

Supplemental information for this article can be found online at http://dx.doi.org/10.7717/peerj-cs.1971#supplemental-information.

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
