# Peer review of "Optimizing type 2 diabetes management: AI-enhanced time series analysis of continuous glucose monitoring data for personalized dietary intervention"

_PeerJ Computer Science, doi:10.7717/peerj-cs.1971_

## Round 0.1 · original submission · Major Revisions

Please consider the reviewer comments.

**Language Note:** The review process has identified that the English language must be improved. PeerJ can provide language editing services - please contact us at [email protected] for pricing (be sure to provide your manuscript number and title). Alternatively, you should make your own arrangements to improve the language quality and provide details in your response letter. – PeerJ Staff

Reviewer 1 ·

Basic reporting

The manuscript entitled “Optimizing type-2 diabetes management: AI-enhanced time series analysis of continuous glucose monitoring data for personalized dietary intervention” has been investigated in detail. The paper discusses the development of an AI-assisted Continuous Glucose Monitoring (CGM) system to help Type-2 Diabetes Mellitus (T2DM) patients improve their health by adjusting their lifestyle. The system involves ML models for predicting blood glucose levels and an AI solution for food recognition. The study claims to assist patients in managing diabetes and bringing HbA1c within the normal range. There are some points that need further clarification and improvement:
1) The paper lacks a clear structure, making it challenging to follow the logic and flow of the content. It needs a well-defined introduction, methodology, results, and conclusion sections.
2) The paper mentions an "all-in-one AI solution for food recognition" without providing details about the solution, its methodology, or its effectiveness. This lack of information raises concerns about the credibility of the proposed system.
3) The dataset used for training ML models is mentioned, but there is a lack of essential details such as the size of the dataset, data collection methods, and patient demographics. A comprehensive description is crucial for evaluating the reliability of the results.
4) The paper mentions Time-Series analysis and forecasting using XGBoost, SARIMA, and Prophet without providing details on how these models are implemented, their parameters, or their comparative performance. Clarity and transparency are essential for assessing the validity of the predictive models.
5) The paper mentions comparing results based on "performance metrics" without specifying the metrics used for evaluation. The lack of clarity on the evaluation criteria undermines the credibility of the comparative analysis.
6) “Discussion” section should be added in a more highlighting, argumentative way. The author should analysis the reason why the tested results is achieved.
7) It will be helpful to the readers if some discussions about insight of the main results are added as Remarks.
This study may be proposed for publication if it is addressed in the specified problems.

The paper lacks a clear structure, making it challenging to follow the logic and flow of the content. It needs a well-defined introduction, methodology, results, and conclusion sections.
The paper mentions an "all-in-one AI solution for food recognition" without providing details about the solution, its methodology, or its effectiveness. This lack of information raises concerns about the credibility of the proposed system.

Experimental design

The dataset used for training ML models is mentioned, but there is a lack of essential details such as the size of the dataset, data collection methods, and patient demographics. A comprehensive description is crucial for evaluating the reliability of the results.

Validity of the findings

The paper mentions Time-Series analysis and forecasting using XGBoost, SARIMA, and Prophet without providing details on how these models are implemented, their parameters, or their comparative performance. Clarity and transparency are essential for assessing the validity of the predictive models.
“Discussion” section should be added in a more highlighting, argumentative way. The author should analysis the reason why the tested results is achieved.

Reviewer 2 ·

Basic reporting

no comments

Experimental design

no comments

Validity of the findings

no comments

Additional comments

The authors consider the important issue of using artificial intelligence to prevent the development of type 2 diabetes. The selected methods and models make it possible to predict the necessary results with sufficient quality and can be favorably evaluated.

---

## Round 0.2 · accepted · Accept

Based on the reviewers' comments, the paper can be accepted.

Reviewer 1 ·

Basic reporting

All my comments have been thoroughly addressed. It is acceptable in the present form.

Experimental design

All my comments have been thoroughly addressed. It is acceptable in the present form.

Validity of the findings

All my comments have been thoroughly addressed. It is acceptable in the present form.

Additional comments

All my comments have been thoroughly addressed. It is acceptable in the present form.